# Caring for Canines: A Survey of Dog Ownership and Parasite Control Practices in Ireland

**DOI:** 10.3390/vetsci10020090

**Published:** 2023-01-24

**Authors:** Caoimhe Sherlock, Celia V. Holland, Jason D. Keegan

**Affiliations:** Department of Zoology, Trinity College, D02 PN40 Dublin 2, Ireland

**Keywords:** dog, dog owner, *Toxocara*, helminth control, fouling

## Abstract

**Simple Summary:**

Dogs are the most common pet chosen by families. Not only are they fun to be around, but they can even benefit our health by encouraging us to exercise when taking them for a walk. They can also reduce our levels of stress and anxiety by acting as a source of comfort and contact. However, living so closely with an animal comes with certain risks. People can catch a number of diseases from pets. These risks can be minimised with proper care and management of our pet dogs. In this study, we aimed to establish if owners were aware of the risks involved in keeping a pet dog and if they were taking the appropriate steps to minimise their own risk of acquiring an illness from their pet. Our results have shown that the majority of owners do not adhere to recommended parasite control practices. Many owners also do not dispose of their dogs’ faeces correctly, further increasing their infection risk. These results indicate that we, as researchers and veterinarians, need to do a better job educating the public about the risks involved in dog ownership and the simple steps that we can take to minimise those risks.

**Abstract:**

Dogs are an important part of life for many people. Dog ownership can confer various health benefits to their owners, but it also presents some risks. In order to establish if dog owners were aware of these risks, an online survey of dog ownership practices was carried out. The survey was open during the month of January 2022 and received 662 responses regarding 850 individual dogs. Overall, 52% of people reported deworming their dog between zero and twice a year, which is unlikely to reduce the risk of infection to humans. The majority of dog owners (71%) reported disposing of faeces correctly; however, when comparing urban and rural environments, 33% of those in rural environments did not dispose of their dogs’ faeces at all, compared with 3% of people in urban locations. People who obtained their dog during the pandemic brought their dog to the vet and dewormed them more frequently than those who obtained their dog before the pandemic. There were no differences in how faeces was disposed of between these groups. These results indicate that we, as researchers and veterinarians, have much work to do in terms of educating dog owners about the ways in which we can reduce the risk of infection to ourselves and our communities.

## 1. Introduction

It has been estimated that there are around 85 million pet dogs in Europe [1]. Pet ownership provides an opportunity to improve human physical health by acting as an incentive to exercise [2]. It can also improve human mental health by reducing stress and anxiety, with pets offering a source of comfort and contact [3]. Although a strong bond between a pet and their owner can have many benefits, this relationship also increases the risk of exposure to zoonotic diseases [4], with many dog owners unaware of the potential pet-associated risks [5].

*Toxocara canis* and *T. cati* are the common roundworm parasites of dogs and cats. If ingested, the eggs of these parasites can pose a health risk to humans, as well as being of veterinary significance [6,7]. Infection with *Toxocara* spp. primarily occurs when people accidently ingest embryonated eggs, which are commonly found in soil [8], but also occasionally in fruit and vegetables [9], and on dog hair [10]. Infection is more common among toddlers and children, due to their lower standards of hygiene and natural hand-to-mouth behaviour [4].

Geographic location, outdoor access, and behaviour while outdoors are also key factors influencing parasitic infection of domestic pets. Dogs that were given access to the outdoors and were taken on regular walks were found to have *Toxocara* eggs on their paws after walking, while their owners also picked up *Toxocara* eggs on their shoes [11]. Coprophagic behaviour displayed by dogs while outdoors can also influence parasite infections and can interfere with the diagnosis of an infection [12]. It was found that dogs that ranged freely outdoors (50–100% of the time), and dogs that were allowed off the lead, had a significantly higher risk of shedding *Toxocara* eggs [13]. The same can be said for scent rolling (rolling in grass and the faeces of other animals) and influencing infection, as it can increase the risk of contamination from the environment [14].

Since helminth eggs are excreted through pet faeces, dog waste disposal likely plays a role in the perpetuation of soil-transmitted helminth (STH) infections. According to a questionnaire carried out by Habluetzel et al. [15] in Italy, the peri domestic environment, i.e., gardens and dog pens, is the most important defecation site in both rural and urban areas. Substantial egg contamination was found in urban areas, such as parks, which can then become a reservoir for *Toxocara* eggs if not cleaned correctly or if dog faeces are not disposed of.

Gaining more information regarding owner behaviour and attitudes towards anthelminthic treatment, and how a dog’s lifestyle can impact *Toxocara* spp. transmission rates is important. Pet ownership has also become increasingly popular during the COVID-19 pandemic. Throughout the pandemic, the purchasing of pets was seen as a means to improve one’s mental and physical health [16], and as a result, there is an increased number of households with pet dogs in Ireland and the UK [17,18]. In this survey, we aimed to establish the attitudes of Irish dog owners towards caring for their dogs, with a particular emphasis on hygiene and parasite control.

## 2. Materials and Methods

### 2.1. Study Design

To investigate dog behaviour and deworming attitudes among pet owners in the Republic of Ireland (ROI), an online survey was conducted using Google Forms. The survey was open from the 4th of January until the 27th of January 2022. The survey could be filled out voluntarily by the participant if they had access to the Google Forms link that was shared via e-mail and social media. Before beginning the survey, the participants read through a brief explanation that provided information on the purpose of the study, what it would entail, and it clarified that all the responses submitted would be anonymous and kept confidential. Participants were able to withdraw at any stage throughout the survey.

The survey was distributed via e-mail and social media (Facebook, WhatsApp, Instagram, and Twitter) to acquire responses from dog owners of a diverse demography. The social media sites were chosen to gather responses from the largest demographic possible, i.e., Instagram posts were directed at young people while Facebook and WhatsApp allowed people who did not have a social media account to share the survey in group chats and with friends. This ensured that the responses represented the ROI in terms of socio-economic status, rural–urban dwelling, age, and gender. The full questionnaire was comprised of 42 questions with both a multiple-choice and paragraph format so the participants could write their own answer where necessary (the full questionnaire is shown in Appendix A). The multiple-choice format allowed each participant to choose the answer most relevant to them from the list of options available (unless a specification was stated, wherein a box was provided for the participant to fill in their own answer). The first section collated the owner’s demographic information (age, gender, how many dogs they have). The following sections collected information surrounding the dog’s demography and health. They also assessed various aspects of the dog’s lifestyle (e.g., diet and walking) as well as pet care (e.g., regarding anthelminthic treatment, veterinary visits, and dog faeces disposal) that were likely to affect *T. canis* exposure and transmission.

### 2.2. Data Analysis

The survey results were recorded in an excel spreadsheet. Some categories within variables were combined to avoid small cell sizes, to simplify data analysis, and to prevent overfitting the model. Any category that had fewer than five responses in a cell was unable to be included in the chi-square analysis, and therefore needed to be combined. The categories combined included, frequency of vet visits (3, 4, and 5 months), faeces disposal (sewer, toilet, and other), and deworming reasons. For the dog’s age, the responses were divided into three categories: “<1 year”, “1–7 years”, and “7+ years” [13]. While questions related to deworming brands, dog sex, and co-habiting with immunocompromised individuals were included in the survey, they were not investigated further and were excluded from analysis, but all responses can be viewed in the full frequency table (see Appendix A). These factors were not analysed further, due to the diversity of responses to the questions of breed and deworming brand making it difficult to form meaningful categories. As some respondents did not answer every question of the questionnaire, the number of valid responses for each variable is reported.

The 13 risk factors linked to *T. canis* infection were investigated in terms of three explanatory variables. They are, environment in which the respondent lived, past infection of the dog or dogs, and whether the dog or dogs was/were obtained pre- or post-pandemic. Individual chi-square tests were run using Minitab (Minitab, v19, State College, PA, USA) to test for the unconditional association between each factor and the variable of interest.

Three multivariate binary logistic regression models were created. The first model predicted how anthelminthic administration, owners’ attitudes towards deworming, and veterinary care differed between urban and rural environments. The second model predicted how dog lifestyle and ownership decisions (e.g., anthelmintic administration and veterinary care) contributed to *T. canis* positive infection rates. The third model predicted how deworming practices and ownership decisions differed between owners that obtained their dog before the pandemic and those that obtained their dog after the pandemic. The potential risk factors that were chosen to form the full model were only those biologically plausible and unconditionally associated with *p* values of less than 0.05.

Subsequently, binary logistic regression analysis was conducted. To select the optimal model, backwards stepwise selection was performed, starting with the full model and removing variables with a *p*-value of <0.1 in a stepwise manner. Prior to running each model, answers containing missing values for the variables of interest were excluded from the analysis.

## 3. Results

### 3.1. Questionnaire Response

A total of 662 owners responded to the survey. These households were home to 850 dogs. The responses to the questionnaire concerning all dogs in each household are shown in Table 1, and the responses specific to individual dogs are shown in Table 2. The majority of the respondents were female, had one dog, and were aged between 18–24. Most people admitted to washing their hands “sometimes” after touching their dog. Most dogs spent the majority of their time indoors, but 94% had access to an enclosed outdoor space. The majority of people said that they disposed of their dogs’ faeces in the correct type of bin (71%) and that they walked their dogs once or twice a day (69%). Most dogs were also allowed off the lead (61%), and did not perform coprophagia (62%), but did perform scent rolling (74%). Most dogs were dewormed once or twice a year (42%). Owners were given five options to select as the reasons for deworming their dogs, with the option to choose all that apply. The most common reason for deworming was “To protect my dog”, comprising 43% of the responses. The second most popular reason was “because the vet told me to” (24%), followed by “To protect myself” (20%), and “Public health reasons” (13%). Only three respondents answered “I do not deworm my dog”.

### 3.2. Statistical Analysis

#### 3.2.1. Urban and Rural Environments

For a number of the investigated variables, statistically significant differences were reported by dog owners who lived in urban and rural environments (Table 3). A higher proportion of people in urban environments obtained their dog during the pandemic (31%) than those in rural locations (24%). In contrast, a higher proportion of owners in rural households obtained a dog for the first time (83%) than those in urban settings (61%). Owners also differed in terms of how frequently they visited the vet, with those in urban environments tending to bring their dogs to the vet more often than those in the countryside. A higher proportion of rural owners reported that that their dog had a previous worm infection (27%) than the urban owners (20%). The biggest differences in deworming frequency were observed for those that treated their animals from 4 to 12 times a year, with 5% more people choosing this strategy in urban settings than those in rural areas. However, the proportion of urban owners that never wormed their dogs was also 5% higher than that of the rural respondents. Rural owners reported higher levels of hand hygiene than their urban counterparts. The disposal of faeces differed significantly with many more owners in rural locations not disposing of their dogs’ faeces (33%) when compared to the urban owners (3%). The dog diets differed, with urban animals receiving more dog food only diets (70% urban vs. 63% rural) but also more raw meat (6% urban vs. 4% rural), while rural dogs received leftovers more frequently (33% rural vs. 24% urban). Finally, a higher proportion of urban dogs had been neutered than those from the countryside. (77% urban vs. 67% rural).

The variables that differed significantly were included in the binary logistic regression model, to determine which factors were the most important in explaining the differences between the rural and urban environments. A summary of the results of the analysis are shown in Table 4, with the full statistical outputs provided in the Appendix A. The disposal of faeces was the most important of all the significant variables, followed by whether the dog was the household’s first dog. Variables that were removed from the model through the stepwise backward selection included if the dog was received before or during the pandemic, the diet, scent-rolling behaviour, and whether the dog had been neutered.

#### 3.2.2. Reported Previous Infection Status

The chi-square analysis identified a number of significant differences between factors in terms of the past infection status reported by the owners (Table 5). A higher proportion of dogs older than 8 years had previously been infected (51%) than had not (38%). For both the 1–7-year age group and those less than a year old, owners reported that their dogs had not had a previous infection more frequently than those that had (53% and 8% vs. 44% and 5%, respectively). Deworming frequencies also differed, with a higher proportion of owners who reported a previous infection, opting to treat their animals between one and four times a year.

The greatest difference in the deworming frequency was observed for those that reported no previous worm infection, with 13% of these owners never treating their animals, compared to just 4% of those owners who had reported a previous infection not treating their dogs for worms. A higher proportion of dogs that were observed scent rolling were also reported to have had a previous worm infection (80%) compared to those dogs that did not (72%). A higher proportion of dogs that were neutered had no previously reported worm infection when compared to those that had (67%). Dog age and deworming frequency were found to be the most important factors contributing to whether or not a dog had a past infection (Table 6).

#### 3.2.3. Dog Obtained before or during the COVID Pandemic

Expected differences in age were observed, with more younger dogs aged 0–7 obtained during the pandemic, and more older dogs present in households since before the pandemic (Table 7). The frequency of vet visits differed significantly, with a higher proportion of dogs obtained during the pandemic visiting the vet more frequently than those that were in the household before the pandemic. Deworming frequencies also tended to be higher in the households that obtained their dog during the pandemic. A higher proportion of owners that received their dog before the pandemic reported washing their hands after touching their dogs all the time (22%), compared to 15% of those that received their dog during the pandemic, although this group also reported that they washed their hands “sometimes” more frequently (73%) than the pre-pandemic owners (63%). Scent rolling was more often reported by pre-pandemic owners (76%) than pandemic owners (68%). A higher proportion of dogs received during the pandemic were given only dog food as their diet (74%), compared to 64% of their pre-pandemic counterparts. Leftovers were more commonly included in the diets of dogs received before the pandemic, while raw meat was marginally more commonly consumed by dogs received during the pandemic (Table 7). The factors that were included in the final model of the binary logistic regression, following the elimination of terms, are shown in Table 8.

## 4. Discussion

The findings of this survey indicate that dog owners are poorly informed regarding appropriate parasite control practices. This is demonstrated by the 52% of respondents who opted to treating their animals for worms once or twice a year, or not at all. Comparable results were seen in a previous European study on 5001 pet owners, where it was found that most dogs (93%) were dewormed considerably less than recommended by the European Consortium for Companion Animal Parasites (ESCCAP) guidelines [19]. ESCCAP recommends treating pet dogs based on the level of risk associated with a dog’s lifestyle [20]. Once or twice a year treatments are recommended for dogs that live predominantly indoors and have little access to the outdoors, other animals, or raw meat. Only 16% of respondents walked their dogs less than daily, indicating that most animals surveyed were regularly outdoors, which results in a level of risk that warrants treatment at least four times a year. The main reason reported by dog owners for deworming their dogs was to protect their pet, with 43% of respondents including this reason in their response. This was followed by “because the vet told me to” comprising 24% of the responses. To protect human health represented 33% of the responses, indicating that a majority of people do not fully understand the motivation for regular anthelmintic treatments, i.e., to eliminate egg shedding in dogs to protect both human and animal health. Similar results were seen in studies conducted across France and Spain, where the deworming behaviour of owners did not match the advised guidelines needed to reduce transmission [21,22]. The ESCCAP guidelines also highlight the importance of regular testing for intestinal parasites by dog owners. We neglected to ask the owners questions about testing their dogs and would recommend that future questionnaires include such a question [23].

Various differences in pet management were also identified in this study. In terms of rural and urban dwelling dogs, the deworming frequency differed somewhat, with a slightly higher proportion of urban dog owners treating their dogs 4–12 times a year. However, a higher proportion of urban dog owners also reported never treating their dogs (13%) compared to rural dog owners (8%). This is of particular concern as the likelihood of infection is higher in dogs that are not dewormed [13]. They also risk shedding eggs in areas more heavily populated with people. More rural dogs were treated four times a year (33%) compared to urban dogs (25%). However, even strict adherence to this regime (i.e., 90% compliance) only results in a reduction in dogs’ egg contribution to the environment from 39% to 28% [24]. It is clear from these results that the public health reasons for deworming our pets need to be emphasized.

A higher proportion of urban dogs visited the vet more frequently than those from rural localities. This may be why rural dogs tend to be more infected than urban dogs, as seen in previous studies [13,25]. Rural owners may visit the vet less frequently due to the relative paucity of veterinary clinics in rural areas compared to urban areas. This agrees with a study carried out in 28 European countries by the Federation of Veterinarians of Europe (FVE), which found that nearly 80% of rural areas across Europe (including Ireland) had a shortage of veterinarians and clinics (FVE, 2020). The shortage of veterinarians in rural regions can have negative effects, where the absence of local veterinary clinics may discourage rural dog owners from travelling long distances to get their dogs checked regularly. The reluctance of rural dog owners to travel long distances, and the remoteness of having a clinic in rural areas may force the closure of rural clinics and cause veterinarians to move to more populated urban areas [26]. This can have serious implications, as a recent study revealed that private practitioners and small veterinary practices are the main sources of advice on animal care for pet owners in Ireland [27].

The results of our model indicated that faeces disposal differed significantly between the urban and rural environments. Owners living in rural areas did not dispose of their dogs’ faeces much more frequently (33%) than owners living in urban areas (3%). Unfortunately, there is a gap in the literature regarding how pet faeces disposal directly influences *Toxocara* spp. transmission. However, the non-disposal of dog faeces can have severe consequences for the surrounding area since animal defecation is the most predominant source of environmental egg contamination [24,28]. Once the eggs become embryonated in the soil, they can survive and remain infectious for years in areas that have frequent human contact, e.g., gardens, parks, and children’s playgrounds [28]. The high rate of non-disposal of faeces in rural areas, along with the lower levels of anthelmintic administration and vet visits, may be reasons why dogs in rural areas can be an important source of infection [25]. A study carried out in Estonia revealed that rural dogs were nine times more likely to be infected with intestinal parasites than urban dogs, while *T. canis* was the most prevalent helminth species in dogs from rural areas of Hungary and the Slovak Republic [25,29,30].

We were also interested in comparing the level of pet care between those who obtained their dog before or during the COVID pandemic. It was hypothesised that there would be a difference in the deworming frequency and pet care between the two cohorts of dog owners, with pre-pandemic dog owners having better deworming regimes, regular vet visits, and frequent faeces disposal. This trend was predicted because of the sudden increase in dog adoption throughout the COVID-19 pandemic [17]. Our results revealed that there was a difference between pre- and post-pandemic dog owners, but the opposite results were generated from what was predicted. Our results showed that a higher percentage of pre-pandemic dog owners did not dispose of their dogs’ faeces (16%) than post-pandemic owners (11%). A greater number of post-pandemic dog owners also managed to visit the vet regularly and deworm their dog in line with ESCCAP guidelines. The reason behind post-pandemic dog owners being more vigilant about anthelmintic administration and overall pet care may be because new owners obtained their dog during a period of “lockdown”, and would have spent more time than usual with their dog, establishing a stronger relationship [17]. The establishment of this relationship has been shown to improve human physiological features and may also enhance human management and dog welfare going forward [31]. However, we do acknowledge that the study took part during a period of lockdown, so some of the recently obtained dogs could have been abandoned following the survey.

The results of this survey provide an interesting snapshot into dog ownership practices in Ireland; however, there are some limitations in terms of the conclusions that can be drawn. The survey was open to the public generally, so it does not represent a completely random sample of dog owners. This is reflected in the fact that 79% of the respondents were women, when we should expect closer to a 50:50 split. Extrapolation of these results to the general population could lead to the over- or under-estimation of the proportions of individuals engaging in certain dog management practices [32]. One point to note regarding women respondents is that in many countries, the advertising of dog products mainly targets women [33], so maybe the emphasis needs to be on canine intestinal parasite control instead of dog care products or food. However, even with some over- or under-estimation of worm control practices, a high proportion of our large sample did not conform to the guidelines, and so improving our messaging to these people remains a clear priority.

Due to the nature of this survey being made available to the public via the internet and social media, the majority of respondents were more likely to be of a younger age [34]. Only 18% of the respondents were families with children, which may be because parents have less time to complete such a survey as they may need to look after young children. Another limitation of the survey is that lower socio-economic homes may not have access to the internet to complete the survey, and so this cohort of people may not be fully represented within the responses. Survey bias was an unavoidable issue, as the survey was shared among zoology students and staff as well as WhatsApp and Facebook groups specifically for dog owners. Since these cohorts tend to be more passionate about animals, and some specifically about dogs, there was a potential bias in the response rate in that they spent longer filling out the survey than other participants. By carrying out a survey to collect our data, we had no control over what answers were given and had to trust that participants gave an honest response to all the survey questions. Respondents may have chosen an answer that did not reflect themselves to “look better”, which could have created some bias. An example of where this bias may have occurred were the responses to “does your dog perform coprophagic behaviour?” and “does your dog perform scent rolling?”. The most popular answers were “never” (64%) or “rarely” (42.3%) in both cases, which was not expected. This may have been because the questionnaire was self-reported. Some owners may not have witnessed coprophagic behaviour or scent rolling even when the dog might have performed it, as most dogs were allowed outside unsupervised (51.8%).

## 5. Conclusions

The results of this study indicate that there are multiple aspects of dog lifestyle, such as pet care, anthelmintic administration, and faeces disposal, which possibly contribute towards *T. canis* transmission, with the geographic location also playing a part. The potential risk factors that have been highlighted in this study should be emphasised by veterinarians, and regular check-ups should be advised. Emphasis should be put on the risks associated with low deworming frequencies, faeces disposal, and rural–urban dwelling, as these factors generated significant associations with positive *T. canis* transmission rates in our logistic regression models.

In addition, the result from this study regarding the effect of deworming frequency on increasing *T. canis* positive rates needs to be investigated further. It has recently been revealed that deworming three or four times a year still does not offer complete protection against endoparasites [35,36]. This again stresses the need for regular faecal examination before treatment to verify that all identified helminths are treated efficiently by the selected anthelmintic product(s).

## Figures and Tables

**Table 1 vetsci-10-00090-t001:** Responses of dog owners to the questions regarding their households.

Question (n = Number of Responses)	Category	n (%)	Question (n = Number of Responses)	Category	n (%)
Owner gender (n = 662)	Female	521 (79)	Home location (n = 662)	Rural	210 (32)
	Male	135 (20)		Sub-urban/Urban	452 (68)
	Non-binary	6 (1)	Do you have an enclosed garden? (n = 662)	No	40 (6)
Number of dogs in household (n = 662)	1	512 (77)		Yes	662 (94)
	2	120 (18)	Is your dog supervised when outdoors in an enclosed space? (n = 662)	No	320 (48)
	3	20 (3)		Yes	342 (52)
	4	10 (2)	Do any other animals occupy the space? (n = 662)	No	518 (78)
Owner age range (n = 662)	18–24	239 (36)		Yes	144 (22)
	25–44	113 (17)	Young children in household (n = 662)	No	543 (82)
	35–44	120 (18)		Yes	119 (18)
	45–54	115 (17)			
	55–64	54 (8)			
	65 +	21 (3)			
Do you wash your hands after touching your dog? (n = 662)	All the time	130 (20)			
	Sometimes	442 (67)			
	Never	90 (14)			

**Table 2 vetsci-10-00090-t002:** Responses of dog owners to questions relating to individual dogs.

Question (n = Number of Responses)	Category	n (%)	Question (n = Number of Responses)	Category	n (%)
How are faeces disposed of?(n = 832)	Bin	587 (71)	Does your dog bring prey times home?	No	787 (93)
	Compost	126 (15)		Yes	62 (7)
	Not disposed of	119 (14)	Diet	Dog food only	570 (67)
How often everyday do you walk your dog? (n = 849)	Once	357 (42)		Diet includes leftovers	235 (28)
	Twice	230 (27)		Diet includes raw meat	44 (5)
	Three times	64 (8)	Do any other animals occupy the space? (n = 662)	No	518 (78)
	Four times	19 (2)		Yes	144 (22)
	Not daily	133 (16)	Where does your dog spend most of the day (n = 847)	Enclosed garden	139 (16)
Is your dog allowed off the lead? (n = 849)	No	335 (39)		Farm/Field	38 (5)
	Yes	514 (61)		Outdoors (not enclosed)	3 (0.4)
Does your dog perform coprophagy?	Never	530 (62)		Indoors	667 (79)
	Rarely	247 (8)	Where did you get your dog? (n = 839)	Breeder	194 (23)
	Often	72 (29)		Farm	79 (9)
Does your dog perform scent rolling?	Never	222 (26)		Online ad	40 (5)
	Rarely	371 (44)		Pet dog litter	254 (30)
	Often	256 (30)		Registered breeder	28 (3)
How frequently do you deworm your dog? (n = 794)	Every 4–12 months	131 (15)		Rescue	244 (29)
	Every 4 months	221 (26)	Is this your first dog?	No	421
	Once or twice a year	354 (42)		Yes	259
	Never	88 (10)	Is your dog neutered?	No	227 (27)
				Yes	622 (73)

**Table 3 vetsci-10-00090-t003:** Chi-squared (X^2^) analysis for factors significantly associated (*p* < 0.05) with dogs in urban or rural environments.

Variable	Category	Urbann (%)	Ruraln (%)	X^2^	*p* Value
Dog joined household	Before pandemic	365 (69)	236 (76)	5.07	0.024
	During pandemic	165 (31)	74 (24)		
Is this your first dog?	No	207 (39)	54 (17)	43.50	<0.001
	Yes	327 (61)	261 (83)		
How frequently do you bring your dog to the vet?	Monthly	45 (8)	12 (4)	28.33	<0.001
	Every 3 months	102 (19)	37 (12)		
	Every 4 months	54 (10)	27 (9)		
	Every 5 months	28 (5)	13 (4)		
	Twice a year	152 (28)	91 (29)		
	Once a year	133 (25)	108 (34)		
	Never	20 (4)	27 (9)		
Has your dog ever had worms?	No	425 (80)	230 (73)	4.85	0.028
	Yes	109 (20)	85 (27)		
How often do you deworm your dog?	Between 4 and 12 times a year	92 (19)	39 (13)	12.55	0.006
	Four times a year	122 (25)	99 (33)		
	Once or twice a year	216 (44)	138 (46)		
	Never	64 (13)	24 (8)		
Do you wash your hands after touching your dog?	All the time	98 (18)	73 (23)	10.72	0.005
	Sometimes	344 (64)	212 (67)		
	Never	92 (17)	30 (10)		
How is excrement (faeces/poo) disposed of?	Bin	437 (84)	150 (48)	166.06	<0.001
	Compost	68 (68)	58 (19)		
	Not disposed of	15 (3)	104 (33)		
Does your dog perform scent rolling?	Never	156 (29)	66 (21)	7.00	0.008
	Sometimes	378 (71)	249 (79)		
Diet	Dog food only	372 (70)	198 (63)	8.97	0.011
	Diet includes leftovers	130 (24)	105 (33)		
	Diet includes raw meat	32 (6)	12 (4)		
Neutered	No	122 (23)	105 (33)	11.12	0.001
	Yes	412 (77)	210 (67)		

**Table 4 vetsci-10-00090-t004:** Binary logistic regression analysis for factors that differed for owners from rural or urban localities.

Source	DF	Contribution	Chi-Square	*p*-Value
Regression	15	24%	167.02	<0.0001
First dog	1	4%	20.75	<0.0001
Vet visit frequency	6	3%	12.13	0.059
Past infection	1	1%	4.27	0.039
Deworming frequency	3	2%	8.89	0.031
Hand hygiene	2	2%	6.12	0.047
Faeces disposal	2	12%	87.07	<0.0001
Error	524	76%		
Total	539	100%		

**Table 5 vetsci-10-00090-t005:** Chi-squared (X^2^) analysis for factors significantly associated (*p* < 0.05) with dogs having a previous worm infection.

Variable	Category	Previously Infected with Worms		X^2^	*p* Value
		No n (%)	Yes n (%)		
Dog age	<1 year	50 (8)	10 (5)	10.91	0.004
	1–7 years	353 (54)	84 (44)		
	8 + years	248 (38)	99 (51)		
Environment	Urban/Sub-urban	425 (65)	109 (56)	4.85	0.028
	Rural	230 (35)	85 (44)		
How often do you deworm your dog?	Between 4 and 12 times a year	102 (17)	29 (16)	12.41	0.006
	Four times a year	160 (26)	61 (34)		
	Once or twice a year	271 (44)	83 (46)		
	Never	90 (13)	8 (4)		
Does your dog perform scent rolling>	Never	184 (28)	38 (20)	5.61	0.018
	Sometimes	471 (72)	156 (80)		
Neutered	No	163 (25)	64 (33)	5.02	0.025
	Yes	492 (75)	130 (67)		

**Table 6 vetsci-10-00090-t006:** Binary logistic regression analysis for factors that differed for owners reporting that their dog had a previous worm infection compared to those that did not have a previous infection.

Source	DF	Contribution	Chi-Square	*p*-Value
Regression	6	4%	24.31	<0.0001
Dog age	2	2%	12.56	0.002
Deworming frequency	3	2%	11.42	0.010
Neutered	1	1%	3.58	0.058
Error	545	96%		
Total	551	100%		

**Table 7 vetsci-10-00090-t007:** Chi-squared (X^2^) analysis for factors significantly associated (*p* < 0.05) with owners getting their dog before or during the pandemic.

Variable	Category	Dog Joined Household		X^2^	*p*-Value
		Before the Pandemic n (%)	During the Pandemic n (%)		
Dog age (n = 853)	<1 year	1 (0)	59 (25)	246.45	<0.001
	1–7 years	274 (46)	160 (67)		
	8 + years	322 (54)	19 (8)		
Environment (n = 840)	Urban/Sub-urban	365 (65)	165 (69)	5.07	0.024
	Rural	236 (39)	74 (31)		
How frequently do you bring your dog to the vet? (n = 840)	Monthly	32 (5)	25 (10)	45.99	<0.001
	Every 3 months	74 (12)	65 (27)		
	Every 4 months	54 (9)	24 (10)		
	Every 5 months	30 (5)	11 (5)		
	Twice a year	178 (30)	61 (26)		
	Once a year	198 (33)	41 (17)		
	Never	35 (6)	12 (5)		
How often do you deworm your dog? (n = 785)	Between 4 and 12 times a year	83 (14)	47 (25)	17.11	0.001
	Four times a year	161 (27)	55 (29)		
	Once or twice a year	277 (46)	75 (40)		
	Never	75 (13)	12 (6)		
Do you wash your hands after touching your dog? (n = 840)	All the time	135 (22)	35 (15)	8.43	0.015
	Sometimes	377 (63)	174 (73)		
	Never	89 (15)	30 (13)		
Does your dog perform scent rolling? (n = 840)	Never	146 (24)	76 (32)	4.96	0.026
	Sometimes	455 (76)	163 (68)		
Diet (n = 840)	Dog food only	386 (64)	176 (74)	11.67	0.003
	Diet includes leftovers	187 (31)	47 (20)		
	Diet includes raw meat	28 (5)	16 (7)		
Neutered (n = 840)	No	124 (21)	102 (43)	42.26	<0.001
	Yes	477 (79)	137 (57)		

**Table 8 vetsci-10-00090-t008:** Binary logistic regression analysis for factors that differed for owners who obtained their dog before or during the pandemic.

Source	DF	Contribution	Chi-Square	*p*-Value
Regression	6	7%	41.71	<0.0001
Deworming frequency	3	2%	13.39	0.004
Hand hygiene	2	2%	13.98	0.001
Neutered	1	2%	14.00	<0.001
Error	545	96%		
Total	551	100%		

## Data Availability

The data is available from the authors on request.

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
