# Peer review of "Caring for Canines: A Survey of Dog Ownership and Parasite Control Practices in Ireland"

_vetsci, 2023, doi:10.3390/vetsci10020090_

Round 1

Reviewer 2 Report

This study analyzes dog owners' approach to treatment for intestinal parasites, taking into account variations that emerged during the COVID-19 lockdown. 

Please consider minor suggestion for Discussion:

Human–dog interactions have a positive effect on human sociality and health. The relationship with dogs helps humans to cope with stress during an emotionally challenging period, such as the COVID-19 pandemic. The psychological characteristics of people who care for dogs were considered in https://doi.org/10.3390/vetsci9030145

in addition, the effect of psychological training and education about dog ethological needs on human helping behavior was also analyzed. Were report that the intervention can improve human physiological features and, consequently, may enhance human management and dog welfare.  

Author Response

Dear Reviewers,

Thank you for taking the time to review this paper. We really appreciate your feedback and expertise, and we will aim to consider your comments and include what you have mentioned in the corrections.

Reviewer 2

Please consider minor suggestion for Discussion:

Human–dog interactions have a positive effect on human sociality and health. The relationship with dogs helps humans to cope with stress during an emotionally challenging period, such as the COVID-19 pandemic. The psychological characteristics of people who care for dogs were considered in https://doi.org/10.3390/vetsci9030145

In addition, the effect of psychological training and education about dog ethological needs on human helping behavior was also analyzed. Were report that the intervention can improve human physiological features and, consequently, may enhance human management and dog welfare.

  • The above citation was included in the discussion (line 314) to highlight the benefits of the human dog relationship and how it can make a positive impact on human behaviour and health, especially during stressful periods such as lock down.

Reviewer 3 Report

Paper summary

This study aimed to explore the ownership practices of pet dog owners in Ireland.  662 dog owners completed an online survey with results focused particularly on preventative care, such as deworming, and management of dog fouling. Overall results have a practical application for pet dog owners and professionals, highlighting potential areas where owner education may be useful.

Comments

The title is rather broad and could potentially be narrowed to summarise the focus of the paper is around dog owner management practices in relation to zoonotic disease transmission. The introduction section clearly summarises literature in this area but does not explore other aspects of dog ownership practice in any depth.

The methods section could be improved by

Line 84: Expand to include which types of social media groups were used, how were they identified and choose

Line 101: Include the minimum response size for a category to be combined

Line 105: Explain why these variables were excluded/not explored

While authors have addressed some the method limitations the paper could be strengthen by expanding the limitation section to more clearly state the impact on the results of sample biases of only using online surveys accessible through social media and self-selection by participants.

The authors report that the results of the study show the importance of vets, professionals and researcher in educating dogs owners. However, the paper would benefits from further unpacking how the results could be used to inform better practice.

Author Response

Dear Reviewers,

Thank you for taking the time to review this paper. We really appreciate your feedback and expertise, and we will aim to consider your comments and include what you have mentioned in the corrections.

Reviewer 3

is rather broad and could potentially be narrowed to summarise the focus of the paper is around dog owner management practices in relation to zoonotic disease transmission. The introduction section clearly summarises literature in this area but does not explore other aspects of dog ownership practice in any depth.

  • We decided to adjust the title to “Caring for Canines: A survey of dog ownership and parasite control practices in Ireland”. We think that this encapsulates what the paper is about and highlights the focus of the paper more.

The methods section could be improved by:

Line 84: Expand to include which types of social media groups were used, how were they identified and choose

  • The types of social media groups used were listed in the methods and how they were identified. A variety of social media groups were used to allow the largest demographic of respondents to be targeted by the survey and to accommodate people who did not have social media accounts, i.e., being able to access the survey through WhatsApp groups.

Include the minimum response size for a category to be combined.

  • The minimum response size for a category had to greater than five responses in a cell. Anything less than five responses could not be included in the chi-square analysis and had to be combined. The categories combined were included in the methods (line 106).

Line 105: Explain why these variables were excluded/not explored – accepted.

While authors have addressed some the method limitations the paper could be strengthen by expanding the limitation section to state the impact more clearly on the results of sample biases of only using online surveys accessible through social media and self-selection by participants. – accepted.

  • A limitations section was included in the discussion (line 332-352) to highlight sample bias that was unavoidable when carrying out such a survey. This section also gives some examples of where bias could have occurred within the survey.

The authors report that the results of the study show the importance of vets, professionals, and researcher in educating dogs owners. However, the paper would benefit from further unpacking how the results could be used to inform better practice.

  • A conclusion section was included in the paper to further explain how our results could be used to inform better practice in terms of public awareness and the responsibilities of veterinarians regarding public education of dog intestinal parasites.